# The Seroepidemiology of a Neglected Zoonotic and Livestock Pathogen in Free-Ranging Bovids: Leptospirosis in African Buffaloes (*Syncerus caffer*)

**DOI:** 10.3390/pathogens10091072

**Published:** 2021-08-24

**Authors:** Wynand Goosen, Mark Hamish Moseley, Tanya Jane Kerr, Andrew Potts, Michele Miller

**Affiliations:** 1DSI-NRF Centre of Excellence for Biomedical Tuberculosis Research, South African Medical Research Council Centre for Tuberculosis Research, Division of Molecular Biology and Human Genetics, Faculty of Medicine and Health Sciences, Stellenbosch University, Cape Town 8000, South Africa; wjgoosen@sun.ac.za (W.G.); tjkerr@sun.ac.za (T.J.K.); miller@sun.ac.za (M.M.); 2School of Biological Sciences, University of Aberdeen, Aberdeen AB24 2TZ, UK; 3Agricultural Research Council—Onderstepoort Veterinary Institute, Private Bag X5, Onderstepoort 0110, South Africa; PottsA@arc.agric.za

**Keywords:** disease ecology, One Health, Microscopic Agglutination Typing, leptospirosis, zoonosis, spillover, Africa

## Abstract

Multi-host pathogens are challenging to control and are responsible for some of the most important diseases of humans, livestock, and wildlife. *Leptospira* spp. are some of the most common multi-host pathogens and represent an important cause of zoonotic infections and livestock productivity loss in the developing world, where contact with wildlife species is common. Although there is increasing evidence that cattle in Africa harbour a broad diversity of *Leptospira* genotypes and serovars, little is known about the epidemiology of these pathogens in wild bovids, such as African buffaloes (*Syncerus caffer*). Using microscopic agglutination testing (MAT) on serum samples collected from free-ranging buffaloes (n = 98) captured in the Hluhluwe-iMfolozi Park (HiP), South Africa, we demonstrated an overall seroprevalence of 21% with seropositivity almost exclusively limited to serovar Tarassovi (serogroup Tarassovi). Moreover, we found no evidence of seropositivity in unweaned calves and showed temporal- or herd-specific variation in exposure risk, and increased probability of seropositivity (OR = 5.44, 95% CI = 1.4–27) in female buffaloes. Together, these findings demonstrate that free-ranging African buffaloes are exposed to *Leptospira* spp. infections, providing insights into the epidemiology of an emerging *Leptospira* serovar in herds with an absence of any disease control and minimal management.

## 1. Introduction

Wildlife can play an important role in the epidemiology of multi-host pathogens of humans and livestock by maintaining pathogens in reservoirs of epidemiologically connected reservoir host populations or environments [1] where traditional control measures, such as vaccination, may be challenging. Moreover, the transmission of pathogens from domestic animals [2] or livestock [3] to wildlife has also been shown to have important implications for the conservation of iconic wildlife species. Therefore, understanding the epidemiology of these pathogens in their wildlife reservoir hosts is critical to mitigating their effects in humans, livestock, and wildlife. 

Globally, *Leptospira* spp. are one of the most common and diverse multi-host pathogens and are an important cause of zoonotic infections and livestock productivity loss. Currently, there are 64 *Leptospira* species described, 38 of which belong to pathogenic or potentially pathogenic subclades [4]. However, two species, *L. interrogans* and *L. borgpetersenii*, are responsible for most pathogenic infections [5]. While rodents are frequently implicated as sources of human infection, there is increasing evidence that, in some contexts, livestock may play a key role as reservoir hosts in Africa [6] where molecular [7] and serological [8] studies have demonstrated a broad diversity of *Leptospira* genotypes and serovars. In livestock, *Leptospira* spp. may be transmitted through venereal transmission [9] or contact with environments contaminated with urine from infected hosts [10]. However, laboratory and genomic studies have demonstrated that the relative importance of environmental and direct host-to-host (e.g., venereal) transmission may differ between *L. interrogans* and *L. borgpetersenii*, respectively [5]. 

While there is increasing evidence that leptospirosis poses a significant public health threat [11] in South Africa and that rodents [12], horses [13], and livestock [8] may play a role in the epidemiology of this neglected zoonosis, there are relatively few studies examining *Leptospira* spp. exposure in wild bovids, such as African buffaloes (*Syncerus caffer*), which may act as wildlife reservoir hosts [14]. Using microscopic agglutination typing (MAT) on serum samples collected from free-ranging buffaloes captured in 2016, 2018, and 2019 in the Hluhluwe-iMfolozi Park (HiP) in KwaZulu-Natal (KZN), South Africa, we aimed to (i) characterise the seroprevalence and serological diversity of *Leptospira* spp. infections, (ii) explore the dynamics in MAT titres in paired samples collected three days apart, and (iii) identify temporal, age, and sex patterns in exposure.

## 2. Results

### 2.1. Seroprevalence and Serological Diversity

While most buffaloes were only sampled once, in 2016, 25 buffaloes were sampled twice at a three-day interval. When including animals that tested positive for Leptospira on either sample in 2016, the MAT results revealed a combined seroprevalence, for all years and age groups, of 21% (21 in 98, 95% CI 14–31%). Evidence for exposure was limited to adult and subadult buffaloes, with 95% (20 in 21) of positive samples reacting to serovar Tarassovi (serogroup Tarassovi) and one adult animal, sampled in 2018, reacting to serovar Szwajizak (serogroup Mini) (Table 1). 

### 2.2. MAT Titre Dynamics

In the 25 paired samples (17 adult, seven subadult, and one calf) collected in 2016, titres varied between samples, despite a relatively short interval (three days) between samples (Figure 1). Although discrepancies between the identification of seropositive animals were noted in 64% (seven out of 11; four adult and three subadult) of paired samples, 71% (five out of seven; three adult and two subadult) of animals that switched between positive and negative results exhibited only a single fold dilution change between titres of 1/100 and negative results (Figure 1), suggesting that MAT lacks sensitivity at low titres. However, two samples did show three- and four-fold titre changes between positive and negative samples.

### 2.3. Temporal, Age, and Sex Patterns in Exposure Risk

Using only the results from the first sample taken in 2016 to facilitate comparisons with other years, where only a single sample was obtained, bivariate analyses demonstrated a significantly lower seroprevalence in calves when compared to subadults or adults (χ^2^ = 6.69, *p* = 0.03, *p* value simulated based on 1 × 10^6^ replicates). Overall seroprevalence appeared to decline from 2016 to 2019 (χ^2^ = 5.11, df = 2, *p* = 0.08), largely due to a reduction in the seroprevalence in female buffaloes (Figure 2). Overall seroprevalence tended to be higher in female compared to male buffaloes (χ^2^ = 2.99, df = 1, *p* = 0.08).

Using a Bayesian multivariate generalised linear model approach, findings confirmed that calves were over 100 times less likely to be infected than adult or subadult animals (OR = 0.01, 95% CI ≤ 0.001–0.37) (Figure 3). By accounting for sex, year, and age, female buffaloes were shown to be over five times more likely to be seropositive than male buffaloes (OR = 5.44, 95% CI = 1.4–27) and animals sampled in 2019 were almost seven times less likely to be infected than animals captured in 2016 and 2018 (OR = 0.15, 95% CI = 0.02–0.71) (Figure 3). 

## 3. Discussion

In free-ranging African buffaloes in South Africa, we identified evidence of exposure to *L. borgpeterseni* serovar Tarassovi with temporal differences in exposure risk, increased exposure risk in female buffaloes, and no evidence of exposure in unweaned calves, providing unique insights into the epidemiology of an important zoonotic and livestock pathogen in the absence of control measures. The seroprevalence noted in this study (21%) is higher than that previously identified in free-ranging African buffaloes (1.7%, seven in 406) in the Kruger National Park (KNP) in South Africa [15] and lower than that (42%, 39 in 92) identified in Uganda [14]. However, the seroprevalence is similar to that (19.4%, 392 in 2021) previously identified in cattle in the same province (KZN) in 2001–2003 [8]. The methodological differences between studies, such as the use of ELISA in Uganda [14] versus MAT in South Africa and heat inactivation of samples from the KNP [15], may account for some of the differences in seroprevalence estimates in buffaloes. Although MAT is considered the gold standard serological assay, it has been shown to lack sensitivity in African contexts [16]. Therefore, the seroprevalence estimates presented in this study represent the apparent seroprevalence, since they were not adjusted for test performance. Furthermore, the discrepancies between results from paired samples suggest that seroprevalence within these herds may be higher than estimated here. 

It is also possible that the difficulties of processing samples under challenging field conditions, the prolonged storage of samples, or the stress of capture may explain the larger (three- and four-fold) variation in MAT titres between paired samples. While MAT may lack sensitivity, particularly at low titres, it is highly specific [16] and allows the presumptive identification of the infecting serovar. Apart from a single animal that reacted to *L. interrogans* serovar Szwajizak (serogroup Mini), all positive samples in this study reacted to *L. borgpetersenii* serovar Tarassovi (serogroup Tarassovi). Serovar Tarassovi is increasingly recognised in a variety of hosts in Africa. For example, it has been detected in cattle in South Africa [8,17] and Uganda [16], horses in South Africa [13], and has historically been detected, at low seroprevalence, in buffaloes in other regions of South Africa [15]. In New Zealand, evidence of exposure to serovar Tarassovi in a broad range of livestock species, and its implication in human cases of leptospirosis [18], suggest that this serovar may be an underrecognized cause of livestock productivity loss and zoonotic disease [19]. 

Seroprevalence decreased between 2016 and 2019, largely driven by a reduction in seroprevalence in female buffaloes from a high of 39% (seven in 18) in 2016 to 11% (two in 18) in 2019. However, the sampling of different herds in each year makes it difficult to disentangle temporal trends from herd-specific differences in exposure risk. In arid or semi-arid areas, the aggregation of wildlife and livestock around water sources may drive the transmission of environmentally transmitted pathogens [20] and *Leptospira* spp. can survive for prolonged periods in surface water, which may act as a source of infection for livestock [10]. Therefore, it is possible that the severe drought conditions experienced in South Africa over the earlier sampling period [21], which had significant impacts on HiP [22], may have resulted in increased environmental contamination due to the increased aggregation of wildlife around scarce water sources and increased utilisation of the rivers upstream of HiP by livestock. An environmental, rather than venereal, route of transmission is supported by the broadly similar infection probability in sexually immature subadults and sexually mature adults. 

After accounting for age and temporal trends in seroprevalence, female buffaloes were nearly five times more likely to be infected than male buffaloes. In large scale studies of domesticated water buffaloes in Thailand (n = 1376) [23] and slaughter cattle in Uganda (n = 500) [16], no effects of sex on seroprevalence were noted. Therefore, our findings may reflect differences in the social structure of free-ranging buffaloes and domesticated bovids. Female buffaloes in HiP are found in large (n = 30–250), mixed-sex herds whereas males are predominantly found in smaller (n = 1–12), all-male sub-herds [24]. Even when found within mixed-sex herds, the majority of males do not spend the entire year in the herd, with the proportion of males in these herds ranging between 5% and 25% in the breeding and non-breeding season, respectively [24]. Therefore, it is possible that the aggregation of female buffaloes in larger herds increases their risk of contact with environments contaminated by infected animals. Alternatively, it is possible that the capture process did not allow for the representative sampling of adult male buffalo. For example, older males that have left the breeding herds may be underrepresented in the captured animals.

While these findings demonstrate clear trends in exposure in free-ranging buffaloes, the relatively small sample size in this study, the lack of contemporaneous seroprevalence data in livestock, and the inability to obtain clinical data, such as abortion and stillbirth incidence, from buffaloes raise further questions regarding cross-species infection, the clinical relevance of these findings for buffaloes, and the epidemiology of this emerging serovar. Moreover, the reliance on serological data and the lack of suitable tissues, such as kidney samples, for the direct detection and molecular typing of *Leptospira* spp. infections in buffaloes suggest that conclusions on *Leptospira* spp. diversity in wild buffaloes should be treated with some caution. To address these limitations, further studies are needed to generate molecular typing data from wildlife and livestock, to identify potential routes and drivers of environmental transmission between livestock and wildlife and to clarify the clinical and zoonotic importance of serovar Tarassovi. However, it is clear that, in the absence of any disease control and minimal management of wild bovids, exposure to *Leptospira* spp. can vary widely either temporally or between herds. This finding has implications for the evaluation of control measures in livestock herds and highlights the need to consider these natural variations in exposure when evaluating the efficacy of control measures. 

## 4. Materials and Methods

Serum samples were obtained from 98 buffaloes from three different herds captured at three different locations within the Hluhluwe-iMfolozi Park (HiP) in 2016, 2018 and 2019 (Figure 4). The ~100,000 ha Hluhluwe-iMfolozi Park is situated in KZN province and is the third largest game reserve in South Africa. It has an African buffalo population of 4544 [25], split into herds that change in number seasonally and in response to resource availability, with mixed sex herds that range between 30 and 250 individuals and all male groups that range between 1 and 12 individuals [24]. Buffaloes are free to range within the confines of the HiP but game fencing largely prevents contact with livestock in surrounding communal farms. Management of the buffalo population is restricted to yearly culls and disease surveillance for bovine tuberculosis (bTB). Capture operations for bTB surveillance [26] frequently capture entire herds, resulting in representative sampling of mixed sex herds present at the time of capture [27]. However, as male animals spend the majority of their time segregated from mixed sex herds [24], sera collected during annual bTB surveillance were selected to provide a broadly similar age and sex profile between years. In 2016, animals were captured and held in temporary holding pens to allow interpretation of single intradermal comparative tuberculin tests (SICTT) [26], allowing for the collection of twenty-five paired serum samples collected at three-day intervals. In 2018 and 2019, single serum samples were obtained from captured buffaloes. Ten millilitres of whole blood were drawn from each animal into serum vacutainer tubes (Fisher Scientific, Suwanee, GA, USA) by venipuncture of the jugular vein. Serum samples were left to clot for 2 h at ambient temperature. Thereafter, sera were centrifuged at 1000× *g* for 15 min, and the supernatant was harvested and stored at −80 ℃ until further analysis. 

Live cultures grown in Ellinghausen–McCullough–Johnson–Harris (EMJH) medium with densities of approximately 2 × 10^8^ leptospires per ml were used as the antigens. An agglutination of 50–100% was taken as a reaction. Titration tests on reacting sera were performed by doubling dilutions of sera using phosphate buffered saline (Sorensen’s buffer), resulting in final dilutions of 1/100 through to 1/3200. The endpoint was defined as the dilution of serum that shows 50% agglutination, leaving 50% free leptospires compared with a control culture diluted 1:2 in Sorensen’s buffer. Results are available in Appendix A.

As all positive samples collected in 2016 only reacted to serovar Tarassovi, animals that tested positive in either of the paired samples were considered positive for the calculation of overall population level seroprevalence. However, to enable comparison between 2016 and years where paired samples were not available, only the result from the first sample from 2016 was used for subsequent statistical analyses. Initial exploratory analyses utilised bivariate Chi square tests to examine associations between seroprevalence and age, sex, and year of sampling. Where low expected values violated assumptions, Chi square statistics were simulated using 1 × 10^6^ replicates.

To quantify exposure risk while accounting for variance associated with other covariates, a generalised linear model (glm) was fitted with an individual’s age, sex, and year of sampling as covariates and exposure status as a Bernoulli response (seropositive/seronegative). The glm was implemented in a Bayesian framework using the rstanarm R package using default (weakly informative) priors. To facilitate comparisons between years, the response variable only included the results from the first sample collected in 2016. 

## Figures and Tables

**Figure 1 pathogens-10-01072-f001:**
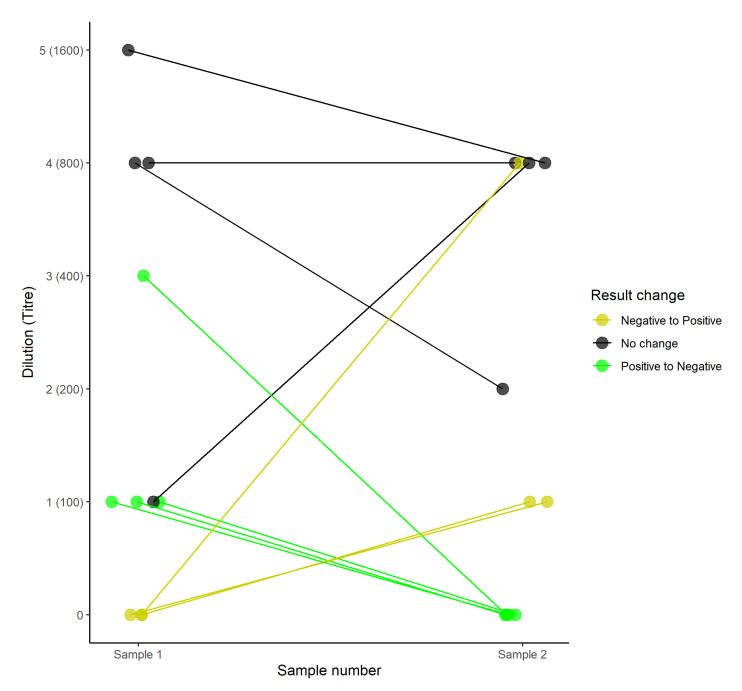
The titre dynamics in paired samples taken three days apart from individual buffaloes in which at least one sample tested seropositive (agglutination at a titre of 1/100). Colours indicate whether there were discrepancies in classification of the animal as seropositive or seronegative.

**Figure 2 pathogens-10-01072-f002:**
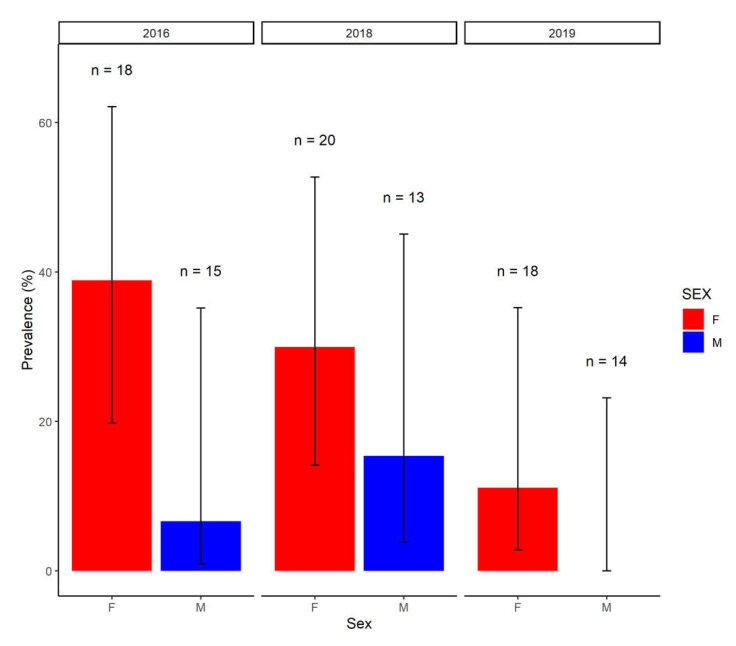
The seroprevalence in male (M) and female (F) buffaloes in each year of sampling. Error bars represent 95% confidence intervals (logit) and the number (n) of male and female buffaloes tested in each year are shown. In 2016, only results from the first sample were considered.

**Figure 3 pathogens-10-01072-f003:**
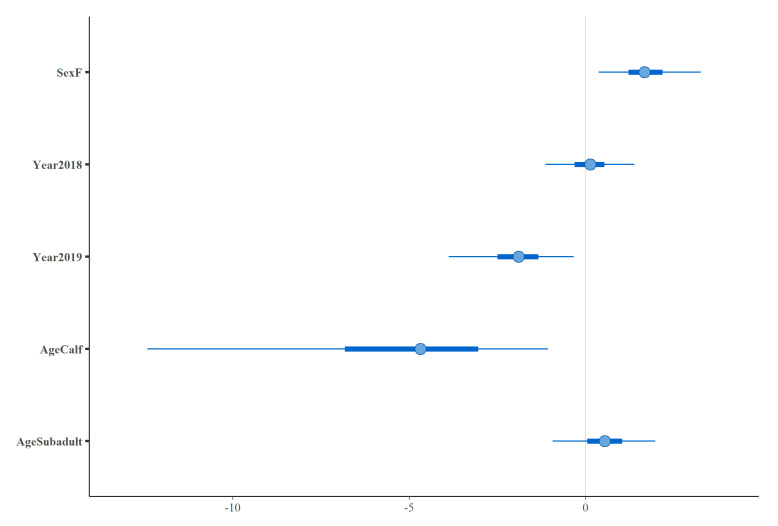
The posterior distributions for temporal (Year), sex (Sex), and age (Age) covariates included in a Bayesian multivariate generalised linear model. Points indicate the posterior mean, thick bars represent 50% confidence intervals and thin bars represent 95% confidence intervals. The reference level for sex was male, 2016 for year of sampling and adult for age.

**Figure 4 pathogens-10-01072-f004:**
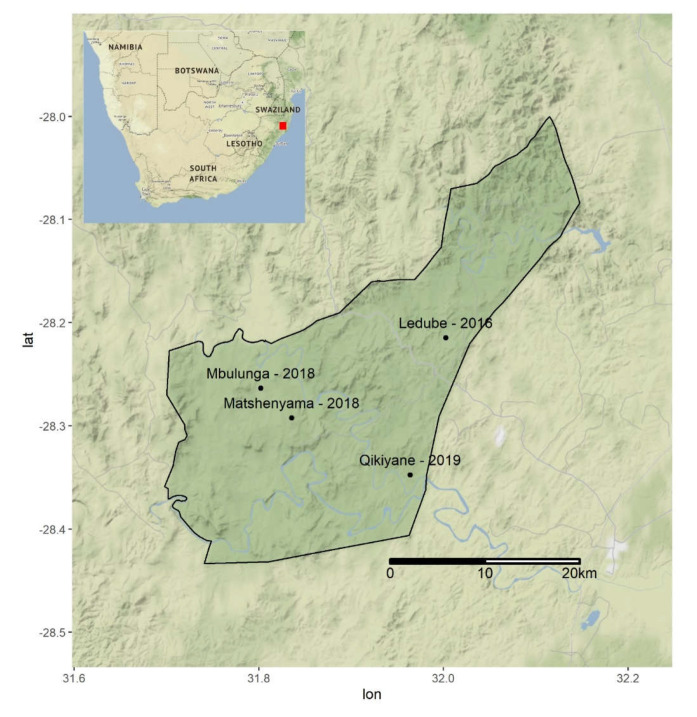
The locations of buffalo sampling efforts in 2016, 2018, and 2019 within the Hluhluwe-iMfolozi Park (HiP). The location of the HiP within South Africa is indicated by the red square in Table 1. All sera were initially screened at a dilution of 1:80 for antibodies against 8 *Leptospira* spp. serovars (8 serogroups) using live antigens (Table 2).

**Table 1 pathogens-10-01072-t001:** Number of animals in each age category positive for the eight Leptospira serovars representing eight serogroups (Australis, Canicola, Icterohaemorrhagiae, Pomona, Tarassovi, Mini, Grippotyphosa, Sejroe) included in the MAT panel across the three sampled years.

		Calf			Subadult			Adult		
		**2016**	**2018**	**2019**	**2016**	**2018**	**2019**	**2016**	**2018**	**2019**
Serovar	Bratislava	0	0	0	0	0	0	0	0	0
	Canicola	0	0	0	0	0	0	0	0	0
	Icterohaemorrhagiae	0	0	0	0	0	0	0	0	0
	Pomona	0	0	0	0	0	0	0	0	0
	Tarassovi	0	0	0	3 *	1	1	8 ^†^	6	1
	Szwajizak	0	0	0	0	0	0	0	1	0
	Grippotyphosa	0	0	0	0	0	0	0	0	0
	Hardjo	0	0	0	0	0	0	0	0	0
Prevalence	Annual (n_positive_/n_tested_) [95% logit CI]	0% (0/5) [0–52%]	0% (0/8) [0–37%]	0% (0/6) [0–46%]	43% (3/7) [14–77%]	25% (1/4) [3–76%]	20% (1/5) [3–69%]	38% (8/21) [20–60%]	33% (7/21) [17–55%]	5% (1/21) [1–27%]
	Overall (n_positive_/n_tested_) [95% logit CI]	0% (0/19) [0–18%]			31% (5/16) [14–57%]			25% (16/63) [16–38%]		

* This value is comprised of animals that tested positive on both paired samples (n = 0), animals that tested positive on sample 1 and negative on sample 2 (n = 2) and animals that tested negative on sample 1 and positive on sample 2 (n = 1). ^†^ This value is comprised of animals that tested positive on both paired samples (n = 4), animals that tested positive on sample 1 and negative on sample 2 (n = 2) and animals that tested negative on sample 1 and positive on sample 2 (n = 2).

**Table 2 pathogens-10-01072-t002:** The *Leptospira* species, serogroups, serovars and strains represented in the MAT antigen panel. These serovars are used in routine diagnosis and include the commonly found strains in various domesticated animal species in South Africa.

Species	Serogroup	Serovar	Strain
*L. interrogans*	Australis	Bratislava	Jez Bratislava
*L. interrogans*	Canicola	Canicola	Hond Utrecht IV
*L. interrogans*	Icterohaemorrhagiae	Icterohaemorrhagiae	RGA
*L. interrogans*	Pomona	Pomona	Pomona
*L. borgpetersenii*	Tarrasovi	Tarrasovi	Perepelitsin
*L. interrogans*	Mini	Szwajizak	Szwajizak
*L. kirschneri*	Grippotyphosa	Grippotyphosa	Moskva V
*L. borgpetersenii*	Sejroe	Hardjo	Hardjoprajitno

## Data Availability

Data are available in Appendix A.

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
