# Peer review of "The Seroepidemiology of a Neglected Zoonotic and Livestock Pathogen in Free-Ranging Bovids: Leptospirosis in African Buffaloes (Syncerus caffer)"

_pathogens, 2021, doi:10.3390/pathogens10091072_

Round 1

Reviewer 1 Report

Dear Authors,

First of all, I would suggest simplifying the title of your manuscript. The titles of the subsections in the Results section should also be changed (2.2 and 2.3), because they do not encourage reading them - the obtained results should be more emphasized. Further, in the introduction, I would like to see a short information about the current main pathogenic Leptospira species, including L. borgpetersenii. It is also worth mentioning the number of females in the appropriate place (for example in the description of Figure 2). I would also like to see the Leptospira reference serovars used in this study (lines 219-222; this text can be converted into a table) in an additional table, which would include the Leptospira species, serogroup and serovar.

Table 1- Leptospira should be in italics.

It is a pity that the serological studies has not been completed molecular analyzes that would give greatly increased value to this work.

Reviewer 2 Report

After reading the publication carefully, I express the opinion that the publication provides valuable knowledge about the Leptospira serovars circulating in the buffaloes population on the African continent. The author relies on samples from bovine tuberculosis testing. I think that the assessment of the occurrence of detected leptospira serovars in terms of sex on the basis of a relatively small number of samples does not fully reflect the actual state. Moreover the author himself presents the research of other authors from Thailand and Uganda where, with a significant greater number of samples tested, no difference in the tested feature was found. Overall, the work is well written and a pleasure to read. I do not have any further comments on the publication, however, it should also be considered whether with a relatively small number of samples tested (seroprevalence assessment), the work should be classified as a case report and not an original article. The statistics was done correctly.

Reviewer 3 Report

This manuscript describes a study of Leptospira seroprevalence in buffaloes from the Hluhluwe-iMfolozi Park in the South African province of KwaZulu-Natal. The study was performed on a limited number of samples, involving 98 animals from 3 herds. On the other hand, the paper can be interesting as a limited knowledge exists for wild bovids and for this particular region. The paper is well-written and easy to read.

General major comments

Internal and external validity should be discussed. I miss information on the demographics of the source population: number of buffaloes (are the 4544 buffaloes all belonging to farms or does the number also include wild free-ranging animals?), geographic distribution, how many herds are in the study area, typical herd size, etc. I miss information on the type and management of farms, I found only a hint at the end of the abstract, but nothing in materials and methods and/or in discussion.

Is the study herds a representative sample? If possible the representativeness of the sample should be evaluated.

The discussion on limitations of the study should be improved, taking into account all sources of potential bias (e.g. sampling, herd representativeness, etc.). Discuss, when possible, both direction and magnitude of any potential bias.

Ethical approval is required (see specific comment lines 198-201).

Specific comments

Line 20: please indicate the number of serum samples collected.

Line 123: please better clarify what is meant by “herd-specific differences”.

Line 126: throughout the study seroprevalence is used, however no adjustments for imperfect test performance has been made, thus the presented results are strictly speaking apparent prevalence. This should at least be mentioned in the discussion.

Lines 137-139: you wrote that the animals, from which the paired samples were collected at three-day interval, were captured and held in temporary holding pens. Then, sera were stored at -80 °C. Therefore, at least the difficulties of processing samples under challenging field conditions and the storage of samples were very similar for paired samples. Even considering possible differences, particularly regarding the stress suffered by the animal for capture, I do not believe “it is likely” that the factors mentioned “can explain” all the observed variations in MAT titres between paired samples, though stress could have a role in these variations. I believe more caution would be appropriate in attributing importance to these factors ("likely" is very strong).

Lines 194-196: how were the herds and the animals selected? Have you defined eligibility criteria for the inclusion of herds in the study (if so, list eligibility criteria)?

Lines 198-201 (major point): a single reference is enough. I suggest keeping reference n. 26, also to avoid an excess of self-citations. However, given that the present study refers to samples other than those of the two studies cited, the ethical approval for this study must be reported in the text. Nevertheless, the data (e.g. prot. n.) relating to the ethical approval of the Stellenbosch University Animal Care and Use committee (or of another committee responsible for animal welfare) must be specified. A generic declaration as in the two studies cited is not enough. Compliance with national and/or international regulations on animal welfare can also be mentioned.

Lines 218-230: in which medium were leptospires grown? Were sera diluted in phosphate buffered saline (PBS)? Were the leptospires added to the diluted sera in the same ratio as in the control with PBS? How was incubation performed? How the results were read?

Round 2

Reviewer 3 Report

Please add the following information, which you have included in the cover letter, also in the manuscript: “Although free to range within the confines of HiP, buffaloes are prevented from leaving the Park boundaries by game fencing. Broadly, animals are not managed beyond yearly culls and disease surveillance for bovine tuberculosis”.

I understand the difficulty in establishing the demographic of the source population and therefore calculating the representativeness of the sample. However, a further indication could be given by the size of the farms tested. Please indicate in the Materials and Methods section the herd size of the three farms tested.

Lines 136-140: it is known that the MAT cannot be standardized and is subject to wide variations in sensitivity, mostly depending on the antigens used in the test and the Leptospira serogroups existing in the region were the animals are found, but also on the test conditions. For these reasons MAT performance is commonly not evaluated and it's okay you didn't. However, your results represent the apparent rather than true seroprevalence not (or not only) because discrepancies between results from paired samples suggest it (as you wrote in the last revision), but because seroprevalence values were not adjusted for test performance (this clearly also applies to other published leptospira prevalence studies). Therefore, I suggest changing the sentence “Moreover, MAT has been shown to lack sensitivity in African contexts (16) and, in this study, the discrepancies between results from paired samples, suggest that these results represent the apparent rather than true seroprevalence and that the true seroprevalence within these herds may be higher than estimated here” to something like (you are free to formulate the sentences differently): "Moreover, MAT has been shown to lack sensitivity in African contexts (16). The seroprevalence values found with the present study represent an apparent prevalence, since they were not adjusted for test performance. Furthermore, the discrepancies between results from paired samples, suggest that seroprevalence within these herds may be higher than estimated here”.

Author Response

Dear Reviewer 3

Thank you for your feedback. We trust that we have addressed your concerns in our responses below and in the attached manuscript. In particular, we would like to thank you for your suggestions regarding the definition of “true” and “apparent” seroprevalence, which we believe highlights our own concerns with the presentation of MAT results. We are happy to address any further comments if necessary.

Kind regards,

Mark Moseley

Reviewer: Please add the following information, which you have included in the cover letter, also in the manuscript: “Although free to range within the confines of HiP, buffaloes are prevented from leaving the Park boundaries by game fencing. Broadly, animals are not managed beyond yearly culls and disease surveillance for bovine tuberculosis”.

Response: The manuscript has been edited to include these statements (line 209-212) in revision 2.

Reviewer: I understand the difficulty in establishing the demographic of the source population and therefore calculating the representativeness of the sample. However, a further indication could be given by the size of the farms tested. Please indicate in the Materials and Methods section the herd size of the three farms tested.

Response: My apologies for the misunderstanding. When we refer to “herds” this does not indicate different farms but rather herds within the larger buffalo population. We have included a new reference to a study of the population biology of the buffalo population in the HiP by Jolles 2007 that complements the previously cited study (Turner et al 2005), which describes the seasonal segregation of male buffalo within an observed herd. Although we do not have data on the size of the herds that were present during capture, Jolles 2007 explicitly states that the age and sex composition of herds captured for bTB surveillance are broadly representative of the mixed sex herds present at the time of capture. The size of these buffalo herds is fluid and larger herds will often split into smaller subgroups periodically (Turner et al 2005). Therefore, the size of mixed sex herds at the time of capture may not be indicative of the size of the herd throughout the year or between years and the only inference we can draw regarding herd size is between female buffaloes, which are always found in larger mixed sex herds, and male buffaloes, which spend most of their time in much smaller all male herds. We have modified lines 207-220 to clarify the fluidity of the herd sizes and to make clear that we are considering herds within a larger buffalo population.

Reviewer: Lines 136-140: it is known that the MAT cannot be standardized and is subject to wide variations in sensitivity, mostly depending on the antigens used in the test and the Leptospira serogroups existing in the region were the animals are found, but also on the test conditions. For these reasons MAT performance is commonly not evaluated and it's okay you didn't. However, your results represent the apparent rather than true seroprevalence not (or not only) because discrepancies between results from paired samples suggest it (as you wrote in the last revision), but because seroprevalence values were not adjusted for test performance (this clearly also applies to other published leptospira prevalence studies). Therefore, I suggest changing the sentence “Moreover, MAT has been shown to lack sensitivity in African contexts (16) and, in this study, the discrepancies between results from paired samples, suggest that these results represent the apparent rather than true seroprevalence and that the true seroprevalence within these herds may be higher than estimated here” to something like (you are free to formulate the sentences differently): "Moreover, MAT has been shown to lack sensitivity in African contexts (16). The seroprevalence values found with the present study represent an apparent prevalence, since they were not adjusted for test performance. Furthermore, the discrepancies between results from paired samples, suggest that seroprevalence within these herds may be higher than estimated here”.

Response: Thank you very much for this suggestion. We have rewritten lines 138-144 to take these suggestions into account. For readers who are not familiar with leptospirosis diagnostics, we have also noted in the manuscript (line 138-139) that MAT is considered the gold standard serological assay for leptospirosis.
